# Systems Analysis for Peptide Systems Chemistry

**DOI:** 10.3390/life9030055

**Published:** 2019-07-01

**Authors:** Martha A. Grover, Ming-Chien Hsieh, David G. Lynn

**Affiliations:** 1School of Chemical & Biomolecular Engineering, Georgia Institute of Technology, Atlanta, GA 30332, USA; 2Department of Chemistry, Emory University, Atlanta, GA 30322, USA

**Keywords:** dynamic chemical networks, origins of life, peptide

## Abstract

Living systems employ both covalent chemistry and physical assembly to achieve complex behaviors. The emerging field of systems chemistry, inspired by these biological systems, attempts to construct and analyze systems that are simpler than biology, while still embodying biological design principles. Due to the multiple phenomena at play, it can be difficult to predict which phenomena will dominate and when. Conversely, there may be no single rate-limiting step, but rather a reaction network that is difficult to intuit from a purely experimental approach. Mathematical modeling can help to sort out these issues, although it can be challenging to build such models, especially for assembly kinetics. Numerical and statistical methods can play an important role to facilitate the synergistic and iterative use of modeling and experiment, and should be part of a systems chemistry curriculum. Three case studies are presented here, from our work in peptide-based systems, to illustrate some of the tools available for model construction, model simulation, and experimental design. Examples are provided in which these tools help to evaluate hypotheses, uncover design principles, and design new experiments.

## 1. Introduction

Biology exploits both physical and chemical phenomena. One approach to understanding biology is to design and construct systems that, while much simpler than biological systems, employ key principles such as selection and feedback. When multiple phenomena occur simultaneously, and at multiple length and time scales, a purely experimental reductionist approach is not well-suited to elucidate competing mechanisms and feedback loops. As the system becomes more complex, a systems approach may be needed instead, in which each physical and chemical event is modeled mathematically, as well as the interactions between components. Once a model of the system has been constructed, it should be validated against experiments. Once validated, it can then be used to make predictions under new conditions and scenarios, to elucidate the key pathways and rate limiting steps, and to design new experiments and systems.

Ideally, the system model would be constructed directly from theory, predicting all covalent and non-covalent interactions and kinetics, and requiring no input from experiments, except for validation of the model. In practice, this ab initio approach is only viable for the simplest of molecular systems due to limitations in computation as well as our understanding. Instead, mathematical models are routinely constructed using experimental data to estimate parameters for chemical rate constants, as well as force fields describing non-covalent molecular interactions. Models of combustion contain thousands of chemical species and reactions [1]. Moreover, models describing both chemical and physical events have been constructed in applications ranging from chemical vapor deposition [2] to peptide catalytic networks [3], and even to whole cells [4].

In this paper, we discuss case studies from our work on peptide-based systems, and how the combination of modeling and experiments has been used to test hypotheses, uncover design principles, and design new experiments. In particular, we highlight numerical and statistical methods that can be valuable to the systems chemist in constructing and interpreting models.

## 2. Materials and Methods

When a reductionist approach is employed to elucidate a single phenomenon, a simple mathematical expression is often used to represent the experimental observations. For example, a rate constant for a first-order reaction may be estimated by measuring the reactant concentration over time. The rate constant may then be estimated from the initial slope. As a second example, this rate constant may be estimated at several temperatures. An Arrhenius model form can be selected to describe the temperature dependence of the rate constant, motivated by transition state theory and having two modeling constants for the prefactor and the observed activation energy. When plotted in the transformed coordinates of lnk and 1/T, the data points fall on a line if the Arrhenius model is a good representation for the data. The model parameters can then be estimated via the slope and intercept of the line.

For these simple models, analytical solutions are possible through algebraic manipulation. Numerical methods are commonly employed for fitting a line to data, with statistical methods for assessing goodness of fit (e.g., R${}^{2}$), and to estimate uncertainty on the parameters using confidence intervals (e.g., t-statistic). As the models become more complex, additional numerical and statistical methods may be needed to execute the steps of model construction, model simulation, and experimental design. Here, we briefly describe the numerical and statistical methods that were used in the case studies, which also have broad applicability to system-level modeling.

### 2.1. Model Construction

The construction of a model is generally not a precise process, but it can be made more systematic using well-established techniques. Two key aspects of model construction are model selection and parameter estimation. Model selection is an open-ended process that may be informed by hypotheses and first-principles models. In practice, model selection may be purely empirical, or alternatively may be a hybrid approach incorporating limited domain knowledge. Once the mathematical form of the model is chosen, then unknown parameters in the model may be estimated to achieve agreement between the experimental data and the model prediction. Additionally, multiple models may be constructed, based on competing hypotheses. After the parameters in each model are estimated, then the best model can be selected based on their relative ability to describe the same dataset.

#### 2.1.1. Parameter Estimation

The goal of parameter estimation is to find the values of unknown parameters, such that the model predictions y(t) match the experimental measurements $y_{m}$. The simplest way to quantify this matching is through the sum-squared error (SSE):(1)$$SSE=\sum_{i=1}^{N_{d}}{\left(y_{m,i}-y\left(t_{i}\right)\right)}^{2}$$
where $N_{d}$ is the number of data points. The goal is then to find the values of the $N_{p}$ model parameters that minimize the sum-squared error. If the model is linear in all unknown parameters, then it is possible to solve for the best-fit parameters using linear algebra, but for many models, especially those involving differential equations, a numerical optimization algorithm such as a gradient search method is required. The best-fit parameters may still not be a good fit, if the model structure is not consistent with the underlying system and data. Chapter 17 of Ref. [5] provides additional description on regression methods.

Once the parameter estimates are obtained, it is also straightforward to calculate confidence intervals on the parameters. A common confidence level used is 95%, and a common distribution is the t-distribution. If the confidence interval on a parameter is large, it may indicate a poor fit (due to a large SSE) or an insufficient number of data points collected. In general, $N_{d}$ should be much larger than $N_{p}$ to avoid overfitting, where the quantity $N_{d}-N_{p}$ is the number of degrees of freedom in the model fit. Chapter 3 of Ref. [6] provides more information about statistical analysis of parameter estimates.

#### 2.1.2. Model Discrimination

It is difficult to know a priori what the mathematical structure of the model should be, thus one approach is to fit multiple model structures to the same dataset, and see which is best. However, by adding more parameters to a model, one can generally lower the SSE, but if there are not enough statistical degrees of freedom, the model will have poor predictive power, due to overfitting. One common criterion to evaluate and compare models is the Akaike Information Criterion (AIC), which balances SSE and $N_{p}$ [7]. A corrected version for small datasets was used in our studies:(2)$$AIC_{c}=N_{d}\ln\frac{\mathrm{SSE}}{N_{d}}+2N_{p}+\frac{2N_{p}\left(N_{p}+1\right)}{N_{d}-N_{p}-1}$$
which is straightforward to evaluate once SSE, $N_{p}$, and $N_{d}$ are known. This AIC value for each model provides a score, such that the model having the best score is selected as the best model for the data. If the amount of data available is limited, then a minimal model may be the best model to describe the system, while if more data are later added, then a more complex model might be selected.

### 2.2. Numerical Simulation of Initial Value Problems

Many models of peptide systems are nonlinear and high dimensional, thus, in general, analytical solution methods are not available and numerical approximate solutions are required. Because we are interested in the time-evolution of the system, the initial conditions are usually specified, and then the system is integrated forward in time, as an initial value problem (IVP) for a system of ordinary differential equations.

#### 2.2.1. Deterministic Simulation

A system of deterministic ordinary differential equations (ODE) described as
(3)$$\frac{dy}{dt}=f\left(t,y\right)$$
with initial condition $y\left(t_{0}\right)=y_{0}$ can be solved approximately by discretizing the time *t* into discrete intervals Δt. Here, the function *f* might consist of chemical rate expressions, with *y* the concentration of one or more species of interest. For the simple irreversible first-order reaction y=C and the reaction rate R=kC, the function is f(t,C)=−R; however, for a reaction network, *y* could be a vector containing multiple species, each reaction *j* would have a rate $R_{j}$, and the functions *f* would contain terms for multiple reactions.

A common class of solution methods for IVPs is the Runge–Kutta method:(4)y[k+1]=y[k]+ϕΔt
where y[k] is the value of *y* at t=kΔt, and time step *k* is an integer. The quantity ϕ is calculated by evaluating the function *f* at one or more points and calculating a typical slope over interval *k*. In the simplest implementation, known as the explicit Euler method, $\phi =f(t_{k},y_{k})$, which is the slope $\frac{dy}{dt}$ at the left end of the time interval. Chapter 25 of Ref. [5] provides additional information on numerical solutions of initial value problems.

#### 2.2.2. Stochastic Simulation

In many molecular systems, stochastic phenomena must be modeled, either because the number of molecules in the system is small, creating large fluctuations, or because the system can take a very large number of discrete configurations, such as the combinatorial sequences a protein can sample. The stochastic simulation algorithm by Gillespie [8] predicts an accurate time evolution for a chemical reaction network. Similar to the deterministic case, the chemical species and the reaction rates *R* are first defined. However, instead of using continuous concentration variables for each species, the number of each molecule is tracked using integer variables.

To simulate one stochastic realization of the system, the chemical events are implemented one at a time. The next reaction to be performed is selected using a random number, but events that have higher rates are selected with proportionally higher probability. The time step between events is also selected using the overall rates with another random number. The random event is selected according to a Poisson distribution, based on the assumption that the individual events are independent.

The stochastic simulation algorithm can be quite slow, especially when there are a large number of molecules and thus a large number of possible events. In the case of stiff systems, where some events are slow and others are fast, the simulation may spend a lot of time executing the fast events, before executing any of the slower events.

### 2.3. Experimental Design

Often a modeling task is undertaken after the data are collected, but ideally the data are collected according to a plan, so that the most knowledge can be obtained from the experimental resources expended [6]. A model can be used to plan the experiments, according to a specific objective. This objective may be to best identify parameters in a particular model, to discriminate between two models (based on two different hypotheses), or to identify the best performance point, such as the maximum yield of a reaction. The experimental conditions in the data collection plan will thus differ for differing objectives. Even if no model exists, the experiments may still be planned, so as to systematically distribute the design points throughout the experimental space.

#### 2.3.1. Model-Free Methods

Factorial design of experiments is commonly used to explore a design space efficiently using limited resources. If a linear model is desired, then high and low levels of each experimental input are selected, but if a maximum is expected to occur inside the design region, then a third level in the middle may also be employed. Even though the sampling along each variable is coarse (only two or three values), the number of experiments required becomes very large when there are a large number of design variables to consider. However, this design of experiments approach does enable understanding of interactions between variables, which is not accessible using the traditional one-factor-at-a-time approach common in scientific studies.

An alternative approach to sampling in high-dimensional spaces is the Latin hypercube method (see Chapter 4 of Ref. [6]). Instead of selecting high and low levels for each experimental input, the number of experiments $N_{d}$ to be performed is first selected. Then, each design variable is discretized into $N_{d}$ intervals. A set of $N_{d}$ experiments is then selected randomly, so that no experiment has the same value of any input. This approach to sampling has been commonly employed for sampling of computer simulations, in which it is often possible to collect larger datasets compared to experimental data collection. Latin hypercube is used in the first case study in this paper, to sample in a high-dimensional parameter space.

#### 2.3.2. Model-Based Methods

None of the studies used in this paper employ model-based experimental design, but common methods include D-optimal design, in which experimental points are designed to minimize the variance (and thus the confidence intervals) on the parameter estimates. Chapter 11 of Ref. [6] provides further description. The D-optimal design objective is also used in Latin hypercube sampling—after calculating many Latin hypercube designs using random numbers, the plan that has the best D-optimal criterion is selected.

Sequential experimental design methods can also be employed in a model-free or a model-based framework. After collecting a limited amount of data, a model can be constructed and used to direct another round of experiments. The response surface methodology typically employs polynomial models [6], but mechanistic models can also be used in the sequential approach [9].

## 3. Case Studies

Three examples from our work on peptide-based assembly and catalysis are briefly described. A particular focus is placed on how numerical and statistical methods promote an effective linkage between modeling and experiment. In these case studies, the numerical and statistical methods are used to test hypotheses, uncover design principles, and design experiments.

### 3.1. Two-Step Nucleation

The folding of proteins is understood to proceed through a molten globule phase, in which the protein is condensed yet disordered. From this condensed state, the protein is then able to fold into its active structure. A similar intermediate state may also be critical for understanding peptide assembly in Alzheimer’s disease. This “two-step” nucleation process has been observed in crystallization via in situ TEM [10]. Nucleation within particles was also observed in peptide assembly of Aβ(16-22), a truncated version of the Alzheimer peptide KLVFFAE [11].

We constructed a model of two-step nucleation for KLVFFAE to better understand the key parameters and environmental conditions controlling the transitions between the solution phase, the disordered particle phase, and the ordered assembly phase (Figure 1) [12]. Here “assembly” refers to a structure with paracrystalline order, which may have either fiber [12,13] or nanotube [14] morphology. A unique aspect of our system, relative to other two-step nucleation systems and models, is that the peptide assemblies nucleate in the particle phase and grow into the solution phase, thus competing with the particle phase for the free peptides in the solution phase.

The model constructed in Ref. [12] is complicated; however, the study elucidated the critical role of solubility in determining the transitions between phases of the system, including the coexistence of phases. The key equations for the solubilities are
(5)$$\begin{array}{ccc} C_{1*} & = & \frac{\rho_{par}e^{-\chi }}{W_{mon}\left(1-e^{-\chi }\right)} \end{array}$$
(6)$$\begin{array}{ccc} C_{2*} & = & \frac{k_{bp}exp\left(\frac{-\Delta G}{k_{B}T}\right)}{k_{g}} \end{array}$$
where $C_{1*}$ is the solubility of peptides from the particle phase, while $C_{2*}$ is the solubility of peptides from the assembly phase. Equation (Equation 5) is based on Flory–Huggins theory, using the thermodynamics of mixing and a lattice assumption. In contrast, Equation (Equation 6) equates kinetic expression for assembly growth and dissolution to predict the equilibrium solubility. The phase that has the greater solubility will thus dissolve into the solution, while the phase with the lesser solubility will incorporate the peptide from solution and grow over time. Key model parameters determining solubility are the Flory–Huggins parameter for the peptide–solvent interaction in solution χ, the binding energy of the peptide assembly ΔG, and the kinetic constant for assembly growth $k_{g}$.

Most parameters in the model were set to nominal representative values for KLVFFAE, such as $\rho_{par}$, the density of the peptide particle, and $W_{mon}$, the peptide molecular weight. However, to explore the range of possible system-level behaviors, as illustrated in Figure 1, 200 parameters sets were selected for χ, ΔG, and $k_{g}$ using Latin hypercube sampling. Each parameter set was simulated using the stochastic simulation algorithm [8] to sample a distribution of particle sizes and assembly lengths, and the long-term behavior of each simulation was classified according to Figure 1. The simulation time was selected to match the length of time for a typical experiment.

Simulations with *C* less than $C_{1*}$ and $C_{2*}$ (“undersaturated”) demonstrated behavior according to Figure 1a. In the supersaturated regime (*C* greater than $C_{1*}$ and/or $C_{2*}$), simulations with $C_{2*}<C_{1*}$ usually demonstrated behavior as in Figure 1g. However, when $C_{2*}>C_{1*}$, then the system usually exhibited behavior such as in Figure 1b. Overall, the simulations showed that the phase behavior could be predicted in most cases based purely on *C*, the initial concentration of peptides in the system, and its relationship to the two solubilities. The phase with the smaller solubility is the observed phase at long times. However, if both phases are undersaturated (*C* is less than $C_{1*}$ and $C_{2*}$), then all peptide remains free in solution. In some cases, the kinetics were too slow to reach steady state during the simulation time, which may also be the case in the experiments.

As shown in Equation (Equation 5), the Flory–Huggins parameter χ impacts $C_{1*}$, but not $C_{2*}$. Simulations showed that particle size increases and particle number decreases with increasing χ [12]. Since χ describes the peptide–solvent interaction, it can thus be tuned experimentally by changing the solvent composition. This interpretation was supported by experiments with KLVFFAE, as shown in the TEM images in Figure 2. As the acetonitrile concentration is increased (i.e., χ is decreased), the number of particles increases and the size of particles decreases. Further experiments with circular dichroism demonstrated that increasing the acetonitrile concentration speeds up peptide assembly, reducing the formation of particles and driving the peptides more rapidly into the assembly phase. Together, the TEM and CD measurements supported the inverse relationship between acetonitrile concentration and Flory–Huggins constant, and the key role of χ in determining the system-level phase behavior.

The use of numerical and statistical tools for simulation (stochastic simulation algorithm) and experimental design (Latin hypercube) enabled the prediction of system-level behavior as a function of three key parameter values. The study provided physical insight into the key role of solubility in determining phase behavior, and suggested a new set of experiments focused on solubility to further support this interpretation.

### 3.2. Selection of Monodisperse Assemblies

The first case study focused on noncovalent interactions, through the assembly of the pre-synthesized KLVFFAE peptide. This second case study includes both covalent and noncovalent interactions [13]. As shown in Figure 3, the tripeptide NFF is modified, such that the carboxylic acid group on F is reduced to an aldehyde, facilitating oligomerization. The imine bond is first formed, followed by cyclization to form N,N-acetal polymers.

The NFF-CHO monomers were first dissolved in solution, at concentrations below their solubility limit. As shown in Figure 4 via HPLC measurements, the monomers first form dimers in solution, with only a small amount of trimer present. At about 6 h, a sudden transition occurred, which was attributed to a phase transition producing particles (and supported by TEM observations). During the next period, 10–60 h, the distribution of polymers shifted toward a mixture of dimers and trimers. The dimer may be partitioned between both the solution and particle phases, while the trimer was assumed to be less soluble, and located primarily within the particle phase. From 10 to 60 h, the size of the particles was also observed to grow in TEM images, even while the oligomer distribution remained constant. At 60 h, a second transition was observed in the oligomer sequence, corresponding to the emergence of peptide assemblies. Notably, as the assembled fibers grew, the oligomer distribution shifted strongly toward the trimer. The degree to which structure in the particle can drive selection is a topic of ongoing investigation, but the assembly of the trimer by the network drives selection.

Each of the three periods in the data was modeled separately with the simple set of reactions:(7)$$\begin{array}{ccc} M+M & ↔ & D \end{array}$$(8)$$\begin{array}{ccc} M+D & ↔ & T \end{array}$$(9)$$\begin{array}{ccc} T & \to & T_{A} \end{array}$$(10)$$\begin{array}{ccc} T+T_{A} & \to & 2T_{A} \end{array}$$
where *M* is the NFF-CHO monomer, *D* is the dimer, *T* is the trimer, and $T_{A}$ is assembled trimer. The final reaction was added based on the hypothesis that assembly is a templating process, and therefore autocatalytic.

The model was simulated with a deterministic algorithm, with parameters estimated according to the values that minimize the sum-squared error. Additionally, various models were estimated using the data in Figure 4, including the full set of four reactions with $N_{P}=6$ rate constants, as well as subsets of the reaction set having fewer reactions with $N_{P}<6$. The Akaike Information Criterion ($AIC_{c}$) was then calculated for each model, and the model including only the first three reactions had the highest AIC score. Thus, the analysis suggested that autocatalysis was not needed to describe the data.

Based on our domain knowledge, we sought to further investigate the autocatalysis hypothesis, and designed a new experiment in which the assemblies were added as seeds into the network. The assemblies grew much faster under the seeded case, providing definitive support for the autocatalysis hypothesis, which could not be supported based only the evidence in Figure 4.

Overall, this combined experimental and modeling study enabled the investigation of a hypothesis in a systematic and quantitative manner. It also demonstrated an interplay between chemical and physical phenomena, with one triggering the other in an alternating fashion, possibly reminiscent of biological emergence.

### 3.3. Catalytic Selection of A Chiral Product

Catalysis is a primary role of proteins in biology. Proteins fold to form enzymes that catalyze chemical reactions with high specificity. While shorter peptides may not fold by themselves, assemblies of multiple peptides can catalyze reactions on their surfaces, similar to enzymes [15]. On the early Earth, such peptide assemblies may have been important catalysts in the origins of life [16].

In Ref. [14], we investigated the ability of KLVFFAL and (Orn)LVFFAL to assemble into nanotubes and to catalyze the cleavage of racemic methodol (β-hydroxyketone) through a retro-aldol reaction. (R)-methodol and (S)-methodol were cleaved with different rates, indicating a specificity reminiscent of enzymes. Several questions arose from these experimental studies, including “What is the nature of the binding site?” and “Is binding or reaction the rate limiting step for the reaction?”

A model was constructed to include both binding and reaction:(11)$$\begin{array}{ccc} E+S↔ & ES\to & E+P_{1}+P_{2} \end{array}$$
(12)$$\begin{array}{ccc} E+R↔ & ER\to & E+P_{1}+P_{2} \end{array}$$
(13)$$\begin{array}{cc} EP_{1}↔ & E+P_{1} \end{array}$$
where *E* is the enzyme, *S* and *R* are (S)-methodol and (R)-methodol, and $P_{1}$ and $P_{2}$ are the products of the reaction.

The model consisted of ordinary differential equations for each species, and was simulated using a deterministic method. Rate constants were estimated by minimizing the sum-squared error. The preliminary results in Figure 5A,B indicate that binding favored (R)-methodol, while reaction favored (S)-methodol. Further experiments were conducted (Figure 5C,D) to demonstrate this intriguing idea—that the selectivity was invertible based on catalyst loading. However, the additional experimental data did not demonstrate this inversion, invalidating the original hypothesis. Because there were not enough data initially, the parameter estimates were not unique (as indicated by large confidence intervals). The larger set of experimental data that was collected, as shown in Figure 5, ultimately led to an improved model with tighter confidence intervals on parameters and a wider range of applicability.

Using the augmented dataset, an AIC analysis supported use of the full model with all four reactions, rather than simplified models based on arguments about rate-limiting steps. Thus, the modeling suggests that both binding and reaction are significant steps for this system, with neither being clearly rate limiting.

In addition to fitting continuous-valued parameters for rate constants, we also needed to specify the number of sites required for each binding event along the surface of the nanotube assembly. In this case, we fit parameters to models having an integer number of peptides constituting the binding sites, in the range of 1–15. The SSE was lowest for a value of 6, increasing significantly for higher and lower values. While we expected the minimal size of the binding site to be 4 based on molecular models, the value of 6 may indicate disorder on the surface, such that all substrate molecules do not tile perfectly in a regular lattice, but rather skip sites along the surface.

In summary, the use of modeling in this catalytic system enabled the investigation of hypotheses, some which were validated and others which were invalidated. The iterative feedback loop between experiments and modeling provides insights into reaction networks that were difficult to measure directly, such as the size of the binding site and the relative importance of binding and reaction.

## 4. Discussion and Conclusions

Life–like behavior in peptide systems may be interpreted in terms of elementary physical and chemical events, e.g., mass–action kinetics, Michaelis–Menton binding, and Flory–Huggins phase behavior. When many events are occurring simultaneously in a network, a model is needed to sort out dominant effects and rate-limiting steps. Numerical methods are usually needed for simulation, and statistical methods can be the impartial arbiter in model selection and parameter estimation. The primary purpose for models should be to direct additional experiments that inform additional modeling, in an iterative feedback loop that allows for the design and construction of more complex systems.

## Figures and Tables

**Figure 1 life-09-00055-f001:**
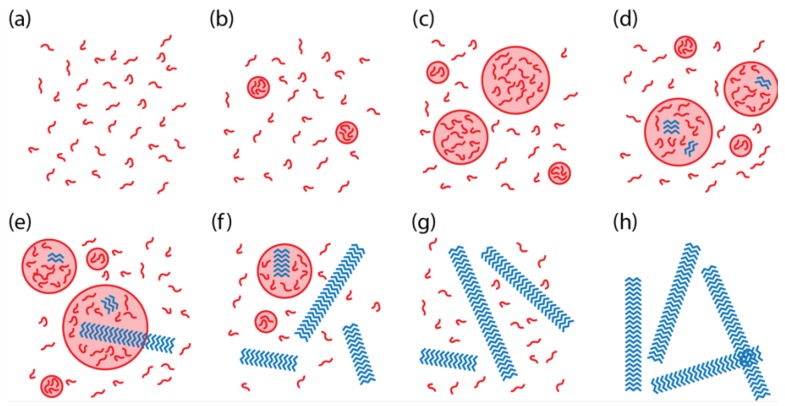
Two-step nucleation in peptide assembly. Peptide assembly mechanism under two-step nucleation. (**a**) Initially, peptides are dissolved in the solution, and (**b**) the peptide particles nucleate if the peptides are not completely soluble. (**c**) The particles grow when the solution is supersaturated for particles. (**d**) Later, the assemblies nucleate inside the particles and (**e**) extend into the solution after growing into a critical size by consuming the peptides inside the particles. (**f**) The assemblies propagate by consuming the free peptides in the solution phase, which decreases the free peptide concentration. (**g**) The particles start to dissolve when the solution becomes undersaturated for particles. (**h**) If assembly dissolution is negligible, after all free peptides are depleted, the assemblies become the only species remaining. Reprinted with permission from Ref. [12]. Copyright (2017) American Chemical Society.

**Figure 2 life-09-00055-f002:**
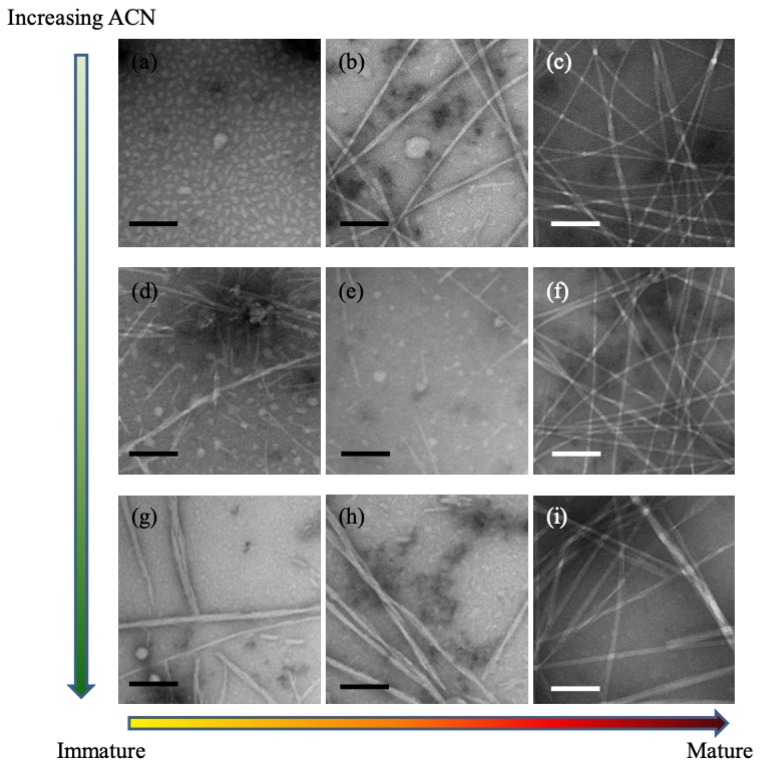
Two-step nucleation experiments. TEM images of 0.5 mM Aβ(16–22) solution in: (**a**–**c**) 40%; (**d**–**f**) 60%; and (**g**–**i**) 80% acetonitrile in water. Images are taken after incubation for: (**a**,**d**,**g**) 1 h; (**b**,**e**,**h**) 5 h; and (**e**,**f**,**i**) 48 h. Scale bar = 100 nm. Reprinted with permission from Ref. [12]. Copyright (2017) American Chemical Society.

**Figure 3 life-09-00055-f003:**
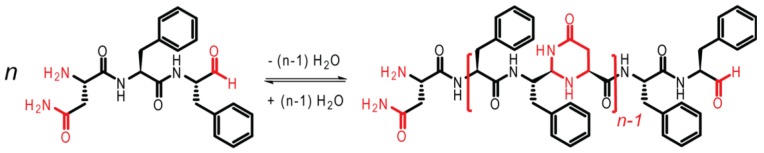
NFF-CHO monomer (**left**) and NFF-CHO oligomers (**right**). Adapted from Ref. [13].

**Figure 4 life-09-00055-f004:**
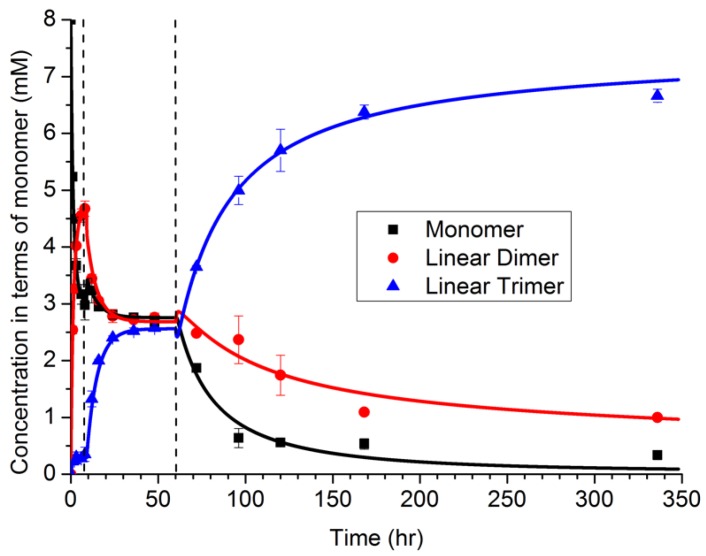
Time progression of the NFF-CHO network. Adapted from Ref. [13].

**Figure 5 life-09-00055-f005:**
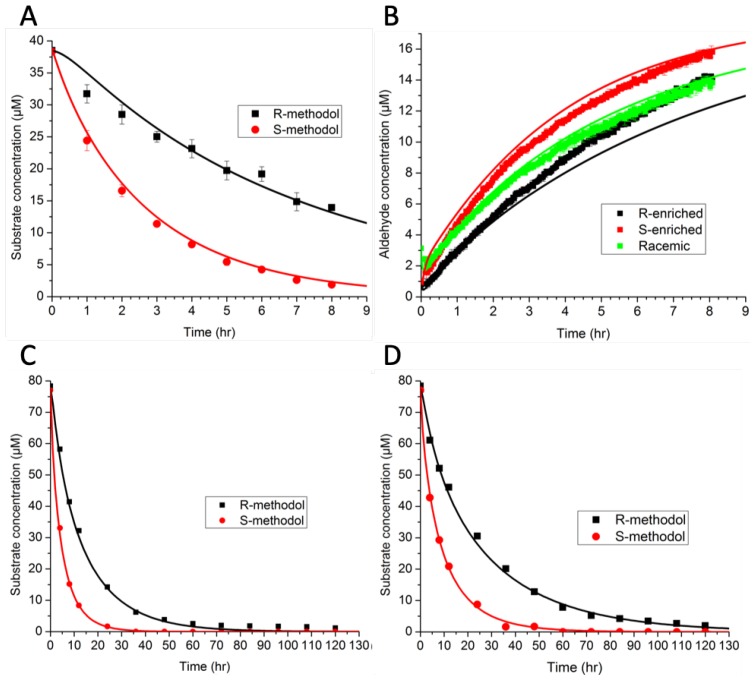
Best fits of Ac-(Orn)LVFFAL-NH2 retro-aldol catalysis with 6 peptides per binding site. (**A**) Chiral HPLC of 500 μM peptide and 76.7 μM (±)-methodol. (**B**) Naphthaldehyde fluorescence of 500 μM peptides with 76.7 μM (±)-methodol (green), 79.2 μM R-enriched methodol (black) and 79.1 μM S-enriched methodol (red). (**C**) Chiral HPLC of 500 μM peptide and 155.3 μM (±)-methodol. (**D**) Chiral HPLC of 300 μM peptide and 155.4 μM (±)-methodol. Solid lines are model fits with: (a) Initial concentrations [E] = 83.3 μM, [S] = 38.3 μM, [R] = 38.5 μM, [P1] = 3.1 μM. (b) Initial concentrations of the racemic solution (green) [E] = 83.3 μM, [S] = 38.3 μM, [R] = 38.5 μM, [P1] = 3.1 μM, for R-enriched substrate (black) [E] = 83.3 μM, [S] = 11.9 μM, [R] = 67.3 μM, [P1] = 0.77 μM and initial concentrations of the S-enriched substrate (red) [E] = 83.3 μM, [S] = 67.2 μM, [R] = 11.9 μM, [P1] = 0.95 μM. (c) The initial concentrations are: [E] = 83.3 μM, [S] = 77.0 μM, [R] = 78.4 μM, [P1] = 4.6 μM. (d) The initial concentrations are: [E] = 50 μM, [S] = 77.0 μM, [R] = 78.4 μM, [P1] = 4.6 μM. Adapted from Ref. [14].

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
