# Peer review of "Systems Analysis for Peptide Systems Chemistry"

_life, 2019, doi:10.3390/life9030055_

Round 1

Reviewer 1 Report

Grover et al., have described three cases of combining experimental data and computational modeling to understand the complex chemical reaction network occurring in the test tube. The data mostly from their previous work (ref 12,13,15) were used and the fitting was performed with the computational models. The computational methods used are the ones used widely in the field and not new. Nevertheless, the manuscript clearly describes what the authors have performed in the current study and their findings. However, I think the manuscript can be improved by adding more explanation about the results as listed below.

For the first case.

Line 198 to 209: The authors describe the reaction dynamics observed with the simulation without no data. It is hard to follow, what they mean by “Analysis of the simulation results demonstrated that the phase behavior could be predicted in most cases based purely on the concentration of peptides in the system C, and its relationship to the two solubilities.” without plots. Similarly, “The modeling study inspired a new set of experiments”, “Simulation results further show that particle size will increase and particle number will decrease with increasing χ”. These sentences also lack the data.

For the third case.

Line 275 to 276: “However, the new experimental data did not demonstrate this inversion, invalidating the hypothesis.” Which data shows the invalidation?

Likewise, the manuscript lacks the details in many places and is more like a review. I personally enjoyed reading the paper, and what they performed are described clearly. However, because of the above reasons, I suggest the authors to add a bit more details before being published.

Reviewer 2 Report

I have read with interest the manuscript presented by Dr. Grover on experiments and modelling of peptide assembly in “particles” and “fibers”, with or without covalent chemical reactions.

The study appears as a follow-up of a previous publication of the same authors (Ref. 12) and thus the first remark is that the Authors should explicitly mention in the abstract, introduction, and conclusions in which respect this study differs from the previous one.

I find the research quite interesting because it deals with a complex and fascinating phenomena, namely phase equilibria of peptides. The Authors also present the “problem” of modelling, giving a good introduction to the topic and assessing the general philosophy of these numerical approach.

Unfortunately no mention was done on “minimal models”. Are these useful? Should researchers focus on minimal models? In which circumstances instead it is better to move toward more complex models? Some sentences commenting “minimal models” in science can be added in order to improve the manuscript.

Another aspect not mentioned is that one of modelling at different “scales” (molecular, supramolecular,…) and how systems spanning multiple scales should be best modelled. I do not intend to invite the Authors to an exhaustive discussion, but a short mention, with references, is probably helpful for most readers.

Suggestion for figure 1: why not adding into the figure the names of the drawn objects? This will help readers in following the discussion of results.

Line 174: it is not clear where the molten-globule conformation play a role when referring, e.g., to Figure 1. Please explain

Equation 6: please define k_bp.

In general, for Equations 5 and 6, please do not limit to show equations, but comment on their significance. Why it should be like that? Explain using intuitive arguments.

Line 208: when particles are mentioned, do the authors refer to the spherical aggregates or elongated aggregates (Figure 1)?

Lines 209 and 211: the Authors are discussing their results but referring one time to ‘chi’ and another time to ACN percentage. As the two terms are actually inversely correlated, the readers get confused. Suggestion, use only one variable (or two variables that go hand in hand, e.g., increase of ACN and decrease of ‘chi’).

Lines 211 and 213: ‘number of particle increases’ and  ‘reducing the formation of particles’. It seems to me that it is not very clearly explained. Please revise.

Line 233: ‘some’? Actually I see from the plot that there is a quite consistent amount of dimer. Please explain

Line 237: are ‘peptide assemblies’ the fiber? I would suggest to give specific names to the two types of aggregating forms, and use them thoroughly in the paper. (Cf. Line 285 ‘nanotube’)

Line 239: ‘but the selection of the trimer by the network drives selection’. Please explain better

Line 240: ‘Each of the three periods in the data was modeled’: It is not immediately clear whether the three periods are modelled “together” or “separately” (one model for each period). Please explain

Caption of Figure 5: ‘Renriched’ should be ‘R-enriched’ in line 3. Moreover the figures on the last 6 lines in Figure 5 caption are best given as table – I guess.

Lines 262-263: is there any evidence (e.g., a plot) for supporting the sentence?

Line 264: Probably most of the readers do not immediately recognize ‘methodol’ as a beta-hydroxyketone, and that the reaction is a retro-aldolic one. Please specify the reaction, show structures, etc.

Lines 271-283: is any data available for displaying? If the article is an experimental report, claims should be accompanied by data. If data are too preliminary, please show only some. This is an important remark that should be seriously taken into account in the revised manuscript. Similarly, if the article is intended as an experimental report, a method section should be added. If experiments are described elsewhere, the method section can be very short, referring – for most details – to previous publications.

Round 2

Reviewer 1 Report

The authors have addressed my main concerns.